# Recovering Ancient Grapevine Cultivars in the Balearic Islands: Sanitary Status Evaluation and Virus Elimination

**DOI:** 10.3390/plants11131754

**Published:** 2022-06-30

**Authors:** Hanan El Aou-ouad, Rafael Montero, Elena Baraza, Josefina Bota

**Affiliations:** 1Research Group on Plant Biology under Mediterranean Conditions (PlantMed), Biology Department, Agro-Environmental and Water Economy Institute-Universitat de les Illes Balears (INAGEA-UIB), Carretera de Valldemossa, km 7.5, 07122 Palma, Spain; hananelaouad@gmail.com (H.E.A.-o.); rafaelmonterosilvestre4@gmail.com (R.M.); elena.baraza@uib.es (E.B.); 2Ecology and Environment Laboratory, Faculty of Science Ben M’sik, University Hassan II, Casablanca 20000, Morocco

**Keywords:** *Vitis vinifera* L., ancient grapevine cultivars, virus incidence, sanitation techniques

## Abstract

Recuperation and genetic diversity preservation of local cultivars have acquired a huge interest in viticulture areas worldwide. In the Balearic Islands, most of the old cultivars are only preserved in grapevine germplasm banks, and so far, the sanitary status of these local cultivars has remained unexplored. The aim of this study was to survey and detect the virus incidence of all conserved cultivars in the government Grapevine Germplasm Bank of the Balearic Islands and to promote the sanitary recovery of two important minor cultivars, Argamussa and Gorgollassa. Enzyme-linked immunosorbent assay (ELISA) screenings were performed on 315 vines of 33 local cultivars. It was shown that the local cultivars were highly infected with simple (39.7%) and mixed infections (52.1%) and only 8.25% of them were free from the viruses tested. Grapevine leafroll-associated virus 3 (GLRaV-3) infection was the most common (82%). Moreover, Grapevine fanleaf virus (GFLV) and Grapevine fleck virus (GFkV) were also present with considerable incidence (25.4% and 43.5%, respectively). In addition, two sanitation protocols were used: shoot tip culture (ST) and thermotherapy in combination with shoot tip culture (CT). Virus elimination using only ST was effective to obtain “healthy” vines of cvs. Argamussa and Gorgollassa. It is important to emphasize that the methods described in the current study were rapid and effective in eliminating both GLRaV-3 and GFLV, also in combination.

## 1. Introduction

Due to several factors, grapevine genetic diversity, gained over the millennia, is being drastically reduced all over the world [1]. At the end of 19th century, different diseases from America reached Europe (powdery mildews, Phylloxera (*Phylloxera vastatrix*)), causing a huge devastation to many European vineyards, which was reflected in drastic changes in the diversity of this species (cultivated and wild grapes). Moreover, over the last 50 years, cultivated grapevines have undergone another drastic reduction owing to the impact of the globalization of wine and the quality demarcation of several cultivars and vineyard areas. In fact, the prevalence of a few cultivars grown worldwide such as Chardonnay, Cabernet Sauvignon, Syrah (Shiraz) and Merlot is increasing and causing, at the same time, the disappearance of old local cultivars that were considered perfectly adapted to the local environmental condition [2,3]. The sanitary selection of disease-free clones has also induced a reduction in the genetic diversity of these major cultivars around the world. Currently, recuperation and preservation of the genetic diversity of local cultivars are considered a challenge in viticulture worldwide and a necessity to counter genetic erosion. In order to stop this loss of genes and genotypes, nearly every wine-growing area has preserved plants in germplasm banks [1,4].

Grapevine diversity reduction due to the above-mentioned factors has been more pronounced in isolated areas such as the Balearic Islands [5], where, despite their small geographic area, the grapevine diversity is considered very high [6]. Balearic wines were well known worldwide for their high quality since Roman times [7]. More recently, several studies have shown the high oenological aptitude of some minor cultivars from these islands [8,9,10]. As reported by García-Muñoz and García-Muñoz et al. [6,11], the sensorial analysis of wines from 18 minor cultivars revealed a quality similar to that of varieties included in Designation of Origin (DO). Today, wine consumers wish to try new products included in the DO; thus, the use of minor cultivars could be a paramount factor to fulfill this gap and to satisfy DO requirements [12]. Among all the cultivars tested, Argamussa and Gorgollassa presented high oenological potential and were more aromatic than others included in the DO [6]. As a consequence, these cultivars could be taken into consideration for the development of new wine market strategies, which could also play an important role in their conservation.

In spite of its importance, the sanitary status of local cultivars has remained neglected and unexplored, leading to the deterioration and loss of certain cultivars in the Mediterranean areas and other countries [10,13,14,15,16]. Remarkably, surveys of several grape-growing regions have revealed the high prevalence of the most widespread grapevine viral diseases, i.e., leafroll, fanleaf, fleck and rugose wood [16,17,18,19,20,21]. It has been shown that local grapevine cultivars usually present a very high incidence of virus infections [22,23,24]. Particularly, in the Balearic Islands, three Majorcan autochthonous grapevine cultivars (Manto Negro, Callet and Moll) were highly infected by grapevine leafroll-associated viruses (GLRaVs). In fact, only 6.4%, 9.6% and 11.5% of Manto Negro, Callet and Moll, respectively, were not infected by these viruses [14]. This viral infection can strongly complicate the clonal selection and authorization of the cultivars, since the EU rules (Directive 2005/43/EC) require that the initial plant material for vegetative propagation is free from Grapevine fanleaf virus (GFLV), Arabis mosaic virus (ArMV), Grapevine fleck virus (GFkV, only for rootstocks), Grapevine leafroll-associated virus 1 (GLRaV-1) and Grapevine leafroll-associated virus 3 (GLRaV-3). In addition, it has been shown that these viruses are able to change leaf morphology, performance and the ampelographic features of the selected vines [25]. 

It is well documented that some viral infections reduce the productive life of plants and provoke severe reductions in yield and quality [26,27,28,29]. Therefore, it becomes crucial to know the prevalence and distribution of grapevine viruses in order to adopt appropriate control measures to preserve genetic resources and to propagate healthy material. In endemically infected areas, sanitation becomes a necessity in the clonal selection process. Common techniques used for virus and viroid sanitation are meristem culture, somatic embryogenesis and shoot tip culture combined or not with thermotherapy and chemotherapy, which show differential success according to the viral species [30,31,32]. For instance, in vitro chemotherapy was used to sanitize vines infected with GFkV [31] and Grapevine rupestris stem pitting-associated virus (GRSPaV) [33]. In order to eliminate GFLV, somatic embryogenesis was used alone [34] or in combination with thermotherapy [35]. This technique was also used to eliminate GLRaV-1, GLRaV-3, GVA and GRSPaV from three grapevine wine cultivars, namely Müller-Thurgau, Grignolino and Bosco [36], and to produce vines free from GLRaVs [37], GFkV [38] and ArMV [39]. To eliminate GFkV from cv. Manto Negro, the combination of meristem or shoot tip culture with thermotherapy (in vivo/in vitro culture) was used [40]. Heat treatment above 35 °C was reported to impede virus replication. Above this temperature, the principal alterations in viral particles were related to the rupture of hydrogen and disulfide bonds of capsid protein, followed by nucleic acid phosphodiester covalent bonds, and consequently to the enhancement of the disorganization and deterioration of viral infectivity, thus leading to the eradication of the virus from the shoot tips [30,41]. Since the number of infected plants is very high among local cultivars, quick and easy methods should be sought in order to sanitize a large amount of material.

The aims of this work are as follows: (i) to assess the current sanitary status of the local cultivars conserved in the government Grapevine Germplasm Bank of the Balearic Islands and (ii) to explore the effectiveness of two sanitation methods to obtain non-infected plants of two well-known minority cultivars (Argamussa and Gorgollassa) from this region and to obtain new licensable material to be commercialized. 

## 2. Materials and Methods

### 2.1. Incidence of Virus Infection 

Plant material: The study was conducted on 315 vines of 33 ancient cultivars (listed in Table 1) from the Balearic Islands (Spain), which were conserved in the government Grapevine Germplasm Bank of the Balearic Islands at the experimental station of Sa Granja in Palma de Mallorca. The plants were 10–15 years old and grafted on 99R rootstock. Cultivars were collected between 2009 and 2014 in order to assess their sanitary status. All the material of the collection was previously identified by combining ampelography, microsatellite analysis and synthesis of historical references of the cultivars [5].

Serological tests: In this study, we tested the most harmful grapevine viruses according to the Commission Directive 2005/43/EC amending the Annexes to Council Directive 68/193/EEC on the marketing of grapevine propagation material. Grapevine leafroll-associated viruses 1 to 3 (GLRaV-1, -2 and -3), Grapevine fanleaf virus (GFLV), Arabis mosaic virus (ArMV) and Grapevine fleck virus (GFkV) were tested in all plants by enzyme-linked immunosorbent assay (ELISA) [42] using commercial kits (Bioreba AG, Reinach, Switzerland). In order to obtain results with as much confidence as possible and minimize false negative responses due to the uneven distribution of viruses in plants, the samples were taken according to IMIDA [43], with regard to the kind of tissue, time of collection and physical conditions. The samples were labeled, placed in plastic bags and stored at 4 °C until assays in the laboratory, which were completed within one week after collection. For GFkV and GFLV detection, the sampling was performed in spring (April-May). Each sample consisted of five shoot basal leaves picked around the vine canopy. In the case of GLRaV-1 and -3 and ArMV, samplings were performed in October. For each leaf, the terminal part of the petiole and the contiguous portion of the limb were excised for the extraction. Crude extracts from three healthy plants were used as negative controls. Samples were considered “positive” when the values of absorbance at 405 nm were at least twice those of the negative controls.

### 2.2. Sanitation of Infected Material 

Plant material: Two minor grapevine cultivars were used for the sanitation procedure, i.e., the white cultivar Argamussa and the red cultivar Gorgollassa. ELISA was used to evaluate the presence of the following viruses both in mother plants and in vines after the sanitation treatments: GFLV, GLRaV-1, GLRaV-3, GLRaV-4 (including GLRaV-4, GLRaV-4 strain 5, GLRaV-4 strain 6 and GLRaV-4 strain 9), ArMV and GFkV. Sample collection and virus testing were performed as described above. 

After the sanitation procedures, the absence of viruses was confirmed using the RT-PCR technique. Specific primers, reported by Osman and Rowani [44], were used to amplify GFLV, GLRaV-1 and GLRaV-3. The presence of GFkV was analyzed using a set of specific primers reported by Osman et al. [45]. Unfortunately, RT-PCR was not performed in the case of GLRaV-4. Total RNA was extracted from 50 mg of phloem scrapings from mature canes or leaves using the Spectrum Plant Total RNA Kit (Sigma-Aldrich) according to the manufacturer’s instructions. RNA purity and concentration were measured at 260/280 nm using a spectrophotometer (NanoDrop-1000, Thermo Scientific, Villebon sur Yvette, France). First-strand cDNA synthesis with a final volume of 20 μL was performed using 500 ng of total RNA, 200 units of recombinant Moloney Murine Leukemia Virus (MuLV) reverse transcriptase (Invitrogen Life Technologies, Inc.), 40 units of RNase inhibitor (RNase out, Invitrogen Life Technologies, Inc.), 0.4 mM of dNTPs and 2 mM of random nonamers (Takara Bio, Inc.). The mixture for reverse transcription (20 μL) was incubated for 50 min at 37 °C and the reaction was inactivated by heating it at 70 °C for 15 min.

PCR analysis was performed using 2 μL of diluted (1:100) cDNA in 25 μL of reaction medium containing PCR buffer (100 mM Tris-HCl, pH 9.2; 750 mM KCl and 35 mM MgCl2), 1 mM of dNTPs, 0.5 mM of each primer and 5 units of Taq polymerase (Takara TaqTM, Takara Bio). Thermocycling was performed as follows: 30 min at 52 °C followed by 35 cycles of 94 °C for 30 s, 58 °C for 45 s and 72 °C for 60 s; final extension at 72 °C for 7 min and storage at −20 °C. Finally, 10 mL of amplification product was electrophoresed on a 2% agarose gel in TBE buffer (90 mM Tris–borate, 2 mM EDTA, pH 8.0), stained with ethidium bromide, visualized on a UV transilluminator and photographed. Positive and negative samples of each virus were included as controls in each test.

Sanitation procedure: Two sanitation techniques were compared to determine their effectiveness on the two grape cultivars Gorgollassa and Argamussa: shoot tip culture (SC) and chamber thermotherapy with shoot tip culture (CT). 

The plants used in both methodologies were obtained by direct rooting of 20 cm dormant canes selected from 8–10 mother plants. Dormant canes were directly rooted in a seedling plate with 40 holes filled with organic substrate and perlite mixture (5:1) and were kept in a greenhouse at 25.0 ± 2 °C by day and 18 ± 2 °C by night. Plants were maintained in the greenhouse until sprouting. A total of 67 plants of Argamussa and 300 plants of Gorgollassa were used. After 6–7 weeks, half of the plants were used for SC treatment and the others for CT treatment. CT treatment consisted of gradually increasing the growth temperature in a growth chamber by 4 °C per week, going from 26 °C/22 °C to 37.5 °C/34 °C (day/night). These conditions were maintained for 40 days in a 16-h photoperiod at a light intensity of 56 μmol m^−2^ s^−1^ provided by white fluorescent tubes (LUMILUX Cool White, L18W/840-Osram). 

In both methodologies, plants with actively growing shoots were stripped of leaves and washed with tap water. Single node segments were disinfected using 70% ethanol for 40 s and soaked in an aqueous solution of 10% sodium hypochlorite with a few drops of Tween 20 for 15 min, shaken and rinsed three times with sterile distilled water to remove sterilizing agents. 

After surface sterilization, shoot tips (1–3 mm) were isolated and cultivated on half-strength Murashige and Skoog’s basal medium including vitamins [46], 2% (*w/v*) sucrose and 0.7% agar supplemented with 2.25 mg/L of 6-benzyladenine (MS-BAP), pH 5.7. Glass test tubes containing 20 mL of medium were used. Cultures were incubated in a Fitoclima S600PLH chamber (Aralab) at 23 °C under a 16-hr photoperiod and a light intensity of 56 μmol m^−2^ s^−1^. Relative humidity (RH) was around 60%. After 6–8 weeks, explants were transferred to rooting media containing basal medium (MS) with half strength of macro- and microelements. 

Transfer to ex vitro conditions: Rooted plantlets measuring 3–5 cm were transferred to sterile soil and maintained in a growth chamber at 25 °C under low light intensity and 90% RH for 15 days. Pots were enclosed in clear polyethylene bags to minimize moisture loss. These potted plantlets were transferred into a greenhouse under higher light intensity and still high RH (70–80%). The RH was gradually decreased to obtain full ex vitro acclimation and active growth. Two months later, the plants were placed outdoors under a shading mesh to protect them from direct sunlight. Plants were grown outdoors in 2-L pots filled with organic substrate and perlite mixture (2:1). Approximately one year after acclimation, when the plants had sufficient lignified material, they were tested for the presence of viruses (Figure 1).

Statistical analyses: To analyze the differences in the sanitary status among cultivars, a nominal logistic model was used with the identity of the virus presented as the response variable using the JMP 10 software package.

To compare the effect of sanitation treatments on virus eradication in Gorgollassa plants, generalized linear models (GLMs) with binomial distribution were fit using the JMP 10 software package. 

## 3. Results

### 3.1. Incidence of Virus Infection 

The results of serological analysis revealed that the local cultivars were infected with all viruses tested in this study (GLRaV-3, GLRaV-1, GFLV and GFkV) except for Arabis mosaic virus (ArMV) (Figure 2 and Table 2).

In general, high infection rates were obtained for all the studied cultivars (Figure 2). Only 8.25% of the tested plants were not infected, and multiple infections were very common (52.1%; Figure 2). In the 33 cultivars studied, GLRaV-3 was the most prevalent virus (82%), alone or combined with other viruses. The second most frequent virus in the collection was GFkV, occurring in 24 cultivars with a total incidence of 43.5%. GFLV was found in 14 local cultivars with a total incidence of 25.4%. Finally, GLRaV-1 was the least common, occurring in four cultivars, with only 3.8% total incidence (Figure 2).

The percentage of single infected plants was 39.7%, mainly represented by infection with GLRaV-3 (80%) (Table 2). In this collection, the most frequent double infections were GLRaV-3 + GFkV (71.4% in 17 cultivars) and GLRaV-3 + GFLV (25.2% in nine cultivars). Simple infection with GFLV was very rare, since it was often found combined with GLRaVs and GFkV in local cultivars (Figure 2). Furthermore, in plants with triple infection, the most common combination was GFLV + GFkV + GLRaV-3 (Figure 2, Table 2). 

The incidence and distribution of these viruses was unequal among the cultivars (L-R χ^2^ = 718.73; *P* < 0.0001 nominal logistic model). Details of the sanitary status of all plants included in the collections are explained in Appendix A and summarized in Table 2. 

Among the 33 cultivars analyzed, only nine non-infected vines were found in the cvs. Malvasia de Banyalbufar, Giró-Ros, Gorgollassa and Mancés de Tibus (Table 2). In some cultivars such as Argamussa, Calop Blanc, Calop Negre and Calop Roig, GLRaV-3 and GFkV were detected in 100% of the plants tested within each cultivar, whereas 100% of the tested Magdalena plants were infected only with GLRaV-3 and 88.8% of the Esperó de Gall plants had a triple infection. 

The occurrence of GLRaV-3 (single or combined) ranged from 6% in Vinater Tinto to 100% in 18 other cultivars (Table 2). Only six cultivars, namely Callet Negrella, Mandó, Vinater Tinto, Manto Negro, Mancés de Capdell and Fernandella, were infected by GFkV, while simple infection with GFLV was found only in Gorgollassa (50%) and Vinater Tinto (11.8%) and that with GLRaV-1 only in Malvasia de Banyalbufar (31.25%). 

### 3.2. Sanitation of Gorgollassa and Argamussa Local Cultivars

Argamussa mother plants were originally infected with GLRaV-3 and GFkV, displaying single and double infections. Gorgollassa showed double (GFLV and GLRaV-4) and triple (GLRaV-3, GFLV and GLRaV-4) infections.

In the of case of cv. Gorgollassa, both sanitation treatments, SC and CT, proved to be equally efficient in eliminating viral infection (no significant differences in infection incidence between treatments L-R χ^2^ = 3.33; *P* = 0.07 GLM binomial). The percentage of sanitized plants did not significantly increase when the heat treatment was also applied, with similar sanitation rates when using SC (76.9%) and CT (86.7%). After sanitation treatments, the plants that remained infected showed only the presence of GLRaV-4.

Unfortunately, in the of case of cv. Argamussa, it was impossible to evaluate the results of CT treatment since no plantlet survived. Shoot tip culture alone resulted in only one GLRaV-3+GFkV-free plant (16.7%), while 66.7% of plantlets resulted GLRaV-3-free but still infected with GFkV.

## 4. Discussion 

The results of the current study, conducted on 33 grapevine cultivars from the government Grapevine Germplasm Bank of the Balearic Islands, revealed a high presence of viruses at different levels of incidence. Nevertheless, the test for Arabis mosaic virus (ArMV) resulted negative for all the samples tested. This result was expected since it was noted that the prevalence of ArMV in Spain was very low [47] and it was never found in the Balearic Islands.

In the current study, the prevalence of GLRaV-3 (single or combined) in minor cultivars (82% of tested plants) was higher than that of 70% found in the three major local varieties cultivated in Mallorca (Manto Negro, Callet and Moll) [15]. Several studies have shown that GLRaV-3 is fairly widespread in Spain [48,49,50] as well as in several Mediterranean viticulture areas [16,18,51] and in the rest of the world [17,20,21,52]. One possible explanation for the high incidence of GLRaV-3 with respect to other GLRaVs could be its higher multiplication efficiency in the host, as reported by Velasco et al. [53]. The high spread of GLRaV-3 in our collection is most likely related to the presence of virus vectors (mealybugs). There is clear evidence of the presence of *Planococcus citri*, a well-known vector of GLRaV-3, in vineyards of the Balearic Islands [51]. Other studies proved that in coastal regions, widespread GLRaV-3 infection was commonly associated to mealybug infestations [21,54] since mealybugs prefer mild warm temperatures and high humidity [55,56].

Similar to the findings of other authors [14], our results revealed that the presence of GFLV and GFkV was high in the local cultivars, at 40% and 70%, respectively. In most cultivars, GFLV was combined with GLRaV-3, while double infection with GFLV + GFkV was observed only in the Vinater Tinto cultivar. Remarkably, our results showed that the incidence of GFkV was higher than that of GFLV. High incidence of GFkV was also observed in other growing areas such as Andalucía (Spain), Istria (Croatia), Iran and Chile [53,57,58,59]. In some viticulture regions, GFkV was the most widespread as compared to GLRaV-3, GLRaV-1 and GFLV [60,61]. GFkV is latent in *Vitis vinifera* and in many grapevine rootstocks. The European certification regulations require the absence of GFkV only in rootstocks and not in *Vitis vinifera* L. (Commission Directive 2005/43/EC amending the Annexes to Council Directive 68/193/EEC). However, the presence of GFkV can affect physiological processes [40]. The high incidence of GFkV reported in this study pointed out the importance of this virus in different cultivars of grapevine, concluding that it would be advisable to include the test of this virus in selection programs [14,62]. Even though no vector of GFkV has yet been identified, this result suggests in situ spread of GFkV by vectors since it is frequently found in combination with GLRaV-3 [14]. Indeed, future epidemiological studies on GFkV are required to improve our knowledge on this virus and its putative vectors. 

Under natural conditions, GFLV is transmitted between grapevines by the parasitic nematode *Xiphinema index* in a non-circulative manner [26]. The relatively low incidence of GFLV in this study is not surprising as it could be explained by the low presence of nematodes in the germplasm collection due to the control treatments against this vector. Similar GFLV incidence was recently observed in a germplasm collection of *Vitis vinifera* L. cv. Tempranillo from la Rioja (northern Spain), although its presence implied a highly deleterious effect on growth, yield and the chemical composition of grape berries [63]. In addition to vector-borne transmission, the high incidence of all viruses observed in the local cultivars could also be explained by the vegetative multiplication of infected stocks, since there are no grapevine nurseries in the Balearic Islands and there is no certified planting material of minor cultivars on the market. 

In general terms, our results revealed that the local cultivars conserved in the germplasm collection plot were highly infected and with more than one virus. Multiple viral infection can result in a more substantially negative effect on fruit quality than single infection [64]. Based on the poor sanitary status of local planting material as well as the low possibility to find virus-free vines in nature, the implementation of clean plant material using biotechnological sanitation techniques is considered fundamental to preserve and recover those cultivars. Indeed, our study has been focused on the elimination of double and triple virus infections in two minor cultivars which were demonstrated to have high agronomic and oenological interest [8,9,10,65].

Different techniques have been applied for virus elimination in order to produce certified material, free of the most damaging viral pathogens [30]. These techniques (i.e., in vitro and/or in vivo thermotherapy, meristem culture and chemotherapy) have shown differential success according to the viral species [32]. 

Our results highlighted the effectiveness of both treatments (ST and CT) in eliminating multiple infections in cv. Gorgollassa. Interestingly, the Gorgollassa plantlets that remained infected after the sanitation processes were found to be infected only by GLRaV-4. Hence, the sanitation processes successfully resulted in the elimination of the most spread virus in the collection (GLRaV-3) as well as GFLV. The high level of GFLV elimination obtained in Gorgollassa was also observed in cvs. “Bidaneh Sefid” and “Shahroodi” in Iran, combining meristem culture and thermotherapy [15]. Similarly, several authors have succeeded in eliminating GFLV as well as viruses of the family *Closteroviridae* by combining thermotherapy and in vitro culture of shoot or meristem [66,67,68]. 

Regeneration of cv. Argamussa plantlets using in vitro culture was difficult. These results should be carefully considered in future studies in order to enhance the regeneration and survival rate of this cultivar. In addition, the sanitation rate obtained for infected Argamussa plants using shoot tip culture was very low (only one GLRaV-3+GFkV-free plant). Remarkably, the non-sanitized plants remained infected only with GFkV (66.7%), thus indicating some difficulties in the sanitation of this virus in this cv. Recently, Bota et al. [40] showed that the combination of either high temperature during summer in the field or growth chamber thermotherapy treatment with shoot tip culture was effective for the elimination of GFkV in cv. Manto Negro (25% and 20%, respectively). It seems that the success of thermotherapy depends not only on the virus species involved but also on the specific interaction between the pathogen and the specific genotype [68]. Indeed, much work needs to be conducted in the case of cv. Argamussa in terms of knowing the sensitivity of GFkV elimination by heat (in vivo/natural) treatment. Moreover, GLRaV-3 is easier to eliminate than GFkV, highlighting the effectiveness of this technique to eliminate GLRaV-3. Similarly, the elimination of GLRaV-1 was obtained with a high efficiency (87.5%) by using micro-shoot tip culture (1 mm) [67]. Remarkably, our results also indicated that all viruses present in both cultivars could be eliminated even from bigger explants such as shoot tips (1–3 mm). The use of larger plant tissues (>1 mm) may allow a compromise between virus elimination and regeneration of vines and also to minimize the risk of somaclonal variation as reported for the somatic embryogenesis method [34,36]. 

## 5. Conclusions

The survey conducted in the government Grapevine Germplasm Bank of the Balearic Islands revealed that the local cultivars were highly infected with viruses, either with simple or mixed infections, with GLRaV-3 being the most common virus followed by GFkV and GFLV. This situation may consolidate the necessity of the application of selection programs for recovering ancient local cultivars and to obtain plants suitable for certification. Indeed, our study has optimized two sanitation protocols for double and triple virus elimination. Our results revealed the successful elimination GLRaV-3 causing Grapevine leafroll disease, the most widespread and economically important grapevine virus disease worldwide, as well as GFLV using the methods reported in the current study. The application of thermotherapy in combination with shoot tip culture seems to be valuable because this method is simple, rapid and may have the possibility to eradicate up to three viruses in grapevines. Remarkably, virus elimination using only shoot tip culture was also effective to obtain virus-free Argamussa and Gorgollassa plants. 

## Figures and Tables

**Figure 1 plants-11-01754-f001:**
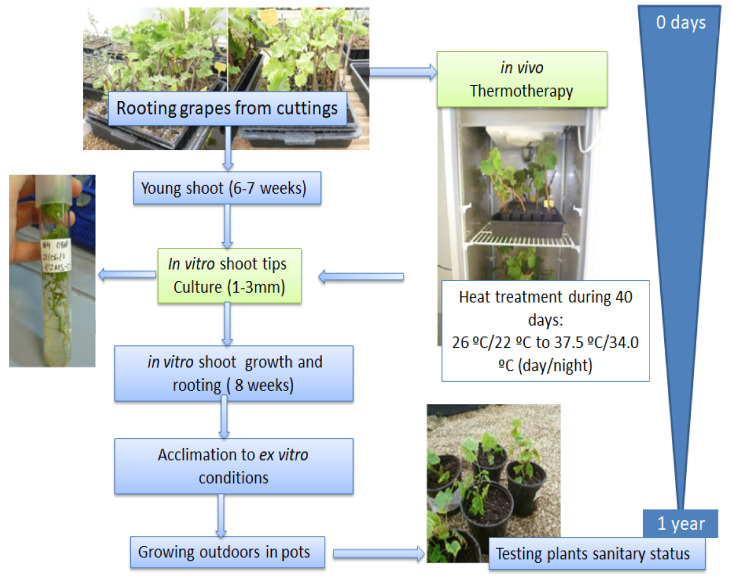
Sanitation procedures from the rooted cuttings to the greenhouse plants in two grapevines cultivars: Argamusa and Gorgollassa.

**Figure 2 plants-11-01754-f002:**
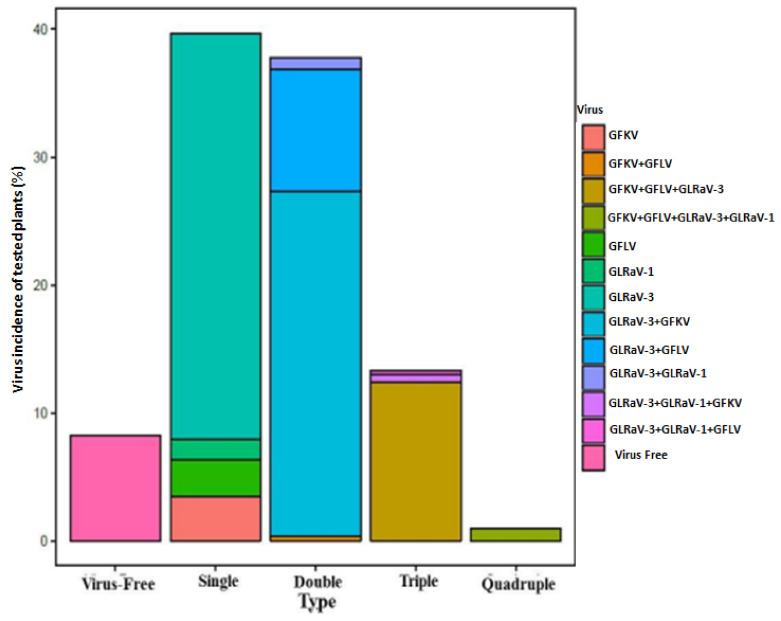
Simple and multiple virus incidences in the cultivars studied (in %). GLRaV-1: Grapevine leafroll-associated virus 1; GLRaV-3: Grapevine leafroll-associated virus 3; GFkV: Grapevine fleck virus; GFLV: Grapevine fanleaf virus.

**Table 1 plants-11-01754-t001:** List of cultivars located in the government Grapevine Germplasm Bank of the Balearic Islands.

Red Grapevine Cultivars	White Grapevine Cultivars
Batista Felanix; Callet Negrella; Calop negre; Callet; Calop Roig; Escursac; Esperó de gall; Fernandella; Fogoneu; Gafarró; Galmater; Gorgollassa; Mancés de Capdell; Mancés de Tibus; Mandó; Manto negro; Sabater; Sinsó; Valent negre; Vinater negro	Argamussa; Calop blanc; Giró Ros; Jaume; Magdalena; Malvasia de Banyalbufar; Mamella de vaca; Mateu; Molinera; Moll; Quigat; Valent blanc; Vinater blanc

**Table 2 plants-11-01754-t002:** Healthy plants (non-infected) and simple, double and multiple virus incidences per cultivar (% of total plants).

Local Cultivars	Plants Number	Non-Infected (%)	Simple Infection (%)	Double Infection (%)	Multiple Infections (%)
Argamussa	9	0	0	100.0 (GLRaV-3+GFkV)	0
Batista Felanix	10	0	77.7 (GLRaV-3)	22.2 (GLRaV-3+GFkV)	
Callet	9	0	77.7 (GLRaV-3)	22.2 (GLRaV-3+GFkV)	
Callet Negrella	7	0	57.1 (GLRaV-3)/67(GFkV)	14 (GLRaV-3+GFkV)	0
Calop Blanc	9	0	0	100 (GLRaV-3+GFkV)	
Calop Negre	9	0	0	100 (GLRaV-3+GFkV)	
Calop Roig	9	0	0	100 (GLRaV-3+GFkV)	
Escursac	10	30	40 (GLRaV-3)	30 (GLRaV-3+GFLV)	
Esperó de Gall	9	0	0	11 (GLRaV-3+GFkV)	88.8 (GLRaV-3+GFkV+GFLV)
Fernandella	9	37.5	37.5 (GLRaV-3)/12.5 (GFkV)	12.5 (GLRaV-3+GFkV)	
Fogoneu	18	0	77.7 (GLRaV-3)	11.1 (GLRaV-3+GFLV)	
Gafarro	7	14.3	85.7 (GLRaV-3)	0	
Galmeter	7	0	0	0	85.7 (GLRaV-3+GFkV+GFLV) 14.2 (GLRaV-3, 1+GFLV+GFkV)
Giró Ros	5	60	40.0 (GLRaV-3)	0	
Gorgollassa	10	10	50.0 (GFLV)	40.0 (GLRaV-3+GFLV)	
Jaumes	8	12.5	12.5 (GLRaV-3)	50.0 (GLRaV-3+GFLV)	25.0 (GLRaV-3+GFkV+GFLV)
Magdalena	7	0	100.0 (GLRaV-3)	0	
Malvasia de B.	16	31.2	12.5 (GLRaV-3)/31.2 (GLRaV-1)	18.7 (GLRaV-3, 1)	6.2 (GLRaV-3+GFkV+GFLV)
Mamella Vaca	3	0	100 (GLRaV-3)	0	
Mancés de Capdell	18	0	44.4 (GLRaV-3)/5.5 (GFkV)	5.5 (GLRaV-3+GFkV)/5.5 (GLRaV-3+GFLV)	38.8 (GLRaV-3+GFkV+GFLV)
Mancés de Tibus	10	70	10 (GLRaV-3)/20.0 (GFLV)	0	
Mandó	8	0	62.5 (GFkV)	37.5 (GLRaV-3+GFkV)	
Manto negro	9	0	55.5 (GLRaV-3)/11.1 (GFkV)	33.3 (GLRaV-3+GFkV)	
Mateu	9	0	100 (GLRaV-3)	0	
Molinera	5	0	100 (GLRaV-3)	0	
Moll	15	0	33.3 (GLRaV-3)	66.6 (GLRaV-3+GFkV)	
Quigat	8	0	0	87.5 (GLRaV-3+GFkV)	12.5 (GLRaV-3+R1+GFkV)
Sabater	9	0	0	77.7 (GLRaV-3+GFLV)	11.1 (GLRaV-3+R1+GFLV) 11.1 (GLRaV-3+GFkV+GFLV)
Sinsó	9	0	78 (GLRaV-3)	22 (GLRaV-3+GFLV)	
Valent Blanc	13	0	0	33.3 (GLRaV-3+GFLV)	50.0 (GLRaV-3+GFkV+GFLV) 8.3 (GLRaV-3, 1+GFLV) 8.3 (GLRaV-3, 1+GFLV+GFkV)
Valent Negre	9	0	0	22.2 (GLRaV-3+GFkV)	66.6 (GLRaV-3+GFkV+GFLV) 11.1 (GLRaV-3, 1+GFLV+GFkV)
Vinater Blanc	9	0	0	100 (GLRaV-3+GFkV)	
Vinater Tinto	17	0	5.8 (GLRaV-3) 5.8 (GFkV) 11.7 (GFLV)	23.5 (GLRaV-3+GFLV) 35.2 (GLRaV-3+GFkV) 5.8 (GFkV+GFLV)	11.7 (GLRaV-3+GFkV+GFLV)

## Data Availability

All data have been included in the main text.

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
