# Peer review of "Recovering Ancient Grapevine Cultivars in the Balearic Islands: Sanitary Status Evaluation and Virus Elimination"

_plants, 2022, doi:10.3390/plants11131754_

Round 1

Reviewer 1 Report

Overall, this manuscript was written in a form that is relatively easy to follow and understandable. The work reported here includes survey for several major grapevine viruses in local grapevine cultivars maintained in the germplasm repository of the Balearic Islands together with the results of an initial attempt to eliminate these viruses from two local cultivars using shoot tip tissue culture and thermotherapy. The topic of this research is of interest both in terms of potential economic utilities of these local cultivars and also in relation to the evolution biology of grapevine viruses in question.  However, this manuscript cannot be published in the present form as several issues need to be addressed. The entire manuscript needs extensive editing to ensure clarity, accuracy and completeness.

Technically, the manuscript suffers from the omission of certain specifics. The survey for viruses relied entirely on ELISA, which may not be reliable depending on the quality of the antibodies used. While the authors stated that RT-PCR was used in the verification tests involving grapevine materials that were subjected to virus elimination, the actual data derived from this line of work was not presented in the paper. Though the authors referred to an earlier publication on the primers used in RT-PCR, the effectiveness and spectrum of these primers in detecting the various strains of each target virus need to be provided. Such information is important when it comes to the reliability of the viral tests given that multiple genetic variants exist for the majority of grapevine viruses. I also feel that the results of virus elimination are questionable given the short time window within which the confirmatory RT-PCR tests were conducted and the very low amounts of templates used in reverse transcription and PCR. Moreover, justification ought to be provided as to why these two cultivars were chosen for virus elimination treatment. Lastly, information should be provided on whether the grapevine cultivars tested had ever been grafted onto any other rootstocks prior to their establishment in the collection. 

Below is a list of additional minor issues that need to be corrected:

Line 49: odd citation.  Change to "As reported by Garcia Munoz et al. (11). Correct other places in the manuscript that have similar issues. 

Line 86-88: unclear what you intend to say here. Rephrase the sentence. 

Table 1: provide the number of plants for each cultivar you included in the test.  This will give the reader an idea on the extent of infection for each of the target viruses you included in the survey. 

Also, change R3 to GLRaV-3 to make things intuitive and consistent with the other viruses included in the table.  

Line 102: change to 'Samples from 3 to 18 ecotypes per cultivar were collected...'

Line 121-122: GFLV and likely other nepoviruses are known to reach  high titres in tissues from the spring. Were there any particular reasons that samples were collected in October instead?  

Line 125: describe what were actually used in your negative controls. 

Line 162: change 'consisted in' to 'consisted of'

Line 181: delete 'and keep culture cabinet conditions'

Line 181: change 'Plantlets' to "These potted plantlets'

Line 187: change "virus presence" to "the presence of viruses". 

Lines 235 and 241: change "In case of" to "In the case of"

Line 240 and elsewhere: change 'GLRaV-4-9' to "GLRaV-4" as GLRaV-9 and other variants are now considered subtypes of the species GLRaV-4.  

Line 239: Not sure how you reached your conclusion here. It seems to me that these two values are very different.  

Line 264: It is unclear how you derived these two values here. Elaborate so that the reader can understand. 

Line 281: change 'from grapevine to grapevine" to "between grapevines"

Line 276-278: Evidence is not convincing. Given the high prevalence of both viruses, it is expected that a large number of plants would be co-infected with both viruses. 

Line 287: change 'grape composition' to 'the chemical composition of grape berries'

Line 302: change 'dangerous' to 'damaging'

Line 322: change "Contrariwise' to "On the contrary'

Line 333: change "big explants' to 'bigger explants'

Line 329: change 'elimination resulted in easier than GFkV' to 'is easier to eliminate than GFkV'

Line 339: insert 'viruses either in' after 'highly infected with'

Supplementary Table S1: Divide the data into two columns, one for GLRaV-1 and the other GLRaV-3.

Author Response

Reviewer #1:

Technically, the manuscript suffers from the omission of certain specifics. The survey for viruses relied entirely on ELISA, which may not be reliable depending on the quality of the antibodies used. While the authors stated that RT-PCR was used in the verification tests involving grapevine materials that were subjected to virus elimination, the actual data derived from this line of work was not presented in the paper.

ELISA provides a reliable diagnosis if samples are collected at the optimal time in the specified vine tissue (Blouin, A.G., Chooi, K.M., Cohen, D., MacDiarmid, R.M. (2017). Serological Methods for the Detection of Major Grapevine Viruses. In: Meng, B., Martelli, G., Golino, D., Fuchs, M. (eds) Grapevine Viruses: Molecular Biology, Diagnostics and Management. Springer, Cham. https://doi.org/10.1007/978-3-319-57706-7_21). For this reason, each virus was tested at the proper moment and in the proper tissue. Additionally, we agree that results from ELISA should be supplemented by molecular tests in critical situations as occurred in a sanitation process. The results from RT-PCR are described in the text (pg 4 lines 144-147)

Though the authors referred to an earlier publication on the primers used in RT-PCR, the effectiveness and spectrum of these primers in detecting the various strains of each target virus need to be provided. Such information is important when it comes to the reliability of the viral tests given that multiple genetic variants exist for the majority of grapevine viruses. I also feel that the results of virus elimination are questionable given the short time window within which the confirmatory RT-PCR tests were conducted and the very low amounts of templates used in reverse transcription and PCR.

The selection of the specific primers was done testing before a broad set of primers reported in the literature using positive and negative controls. Reference of the selected pair of primers for each virus is now reported in the M&M section.

Moreover, justification ought to be provided as to why these two cultivars were chosen for virus elimination treatment.

The recovery of old grapevine material not only contributes to preserve biodiversity but is a source of solutions to deal with the countless challenges that winegrowers face. The recovery of these two pre-phylloxera cultivars may ensure the use of traditional material rather than the massive planting of clones of widespread cultivars. The new reality of the sector is demanding traditional cultivars. The studies about the oenological potential of grapevine cultivars included in the germplasm bank reported very good results for these two cultivars (Garcia-Muñoz, 2011 (PhD thesis)). These two Minor varieties excluded from Quality Demarcations appear to be more aromatic and, as a consequence, Argamusa, Gorgollassa varieties could be taken into consideration for the development of new wine market strategies, which could also play an important role in the conservation of these cultivars.

A deeper explanation regarding the relevance of these two cultivars and the importance for the sector to recover this material was included in the introduction section (pg 2, lines 54-58)

Lastly, information should be provided on whether the grapevine cultivars tested had ever been grafted onto any other rootstocks prior to their establishment in the collection.

As they are pre-Phylloxera cultivars, the original material sent to the national germplasm bank, from which these cultivars were later recovered, came from ungrafted field plants. Once in the germplasm bank, it was grafted for no other purpose than conservation. In recent years, a study of the behaviour of the cultivar Argamusa on different rootstocks (R-110, 140, Rugierei, 1103-Paulsen and SO4) has been carried out in order to see the effects of the rootstock/cultivar combination on the aromatic profile and the oenological vocation, but to the best of our knowledge, the results have not yet been published.

Below is a list of additional minor issues that need to be corrected:

Line 49: odd citation.  Change to As reported by Garcia Munoz et al. (11). Correct other places in the manuscript that have similar issues. 

It’s done. The authors of the references were added according to the referee’s comment.

Line 86-88: unclear what you intend to say here. Rephrase the sentence. 

We rephrased the sentence

Table 1: provide the number of plants for each cultivar you included in the test.  This will give the reader an idea on the extent of infection for each of the target viruses you included in the survey. 

The number of plants is now provided in Table 2

Also, change R3 to GLRaV-3 to make things intuitive and consistent with the other viruses included in the table.  

It has been changed.

Line 102: change ‘Samples from 3 to 18 ecotypes’ per ‘cultivar were collected...’

It has been corrected.

Line 121-122: GFLV and likely other nepoviruses are known to reach high titres in tissues from the spring. Were there any particular reasons that samples were collected in October instead?  

You are right. It was a typographical error. We rectified this information in the M&M section.

Line 125: describe what were actually used in your negative controls. 

The description is added to M&M section: Crude extracts from three healthy plants were used as negative controls.

Line 162: change ‘consisted’ to ‘consisted of’

It has been corrected.

Line 181: delete ‘and keep culture cabinet conditions’

It has been corrected.

Line 181: change ‘Plantlets’ to ‘These potted plantlets’

It has been corrected.

Line 187: change ‘virus presence’ to ‘the presence of viruses’. 

It has been corrected.

Lines 235 and 241: change In case of’ to ‘In the case of’

It has been corrected.

Line 240 and elsewhere: change GLRaV-4-9; to GLRaV-4 as GLRaV-9 and other variants are now considered subtypes of the species GLRaV-4.  

It has been corrected.

Line 239: Not sure how you reached your conclusion here. It seems to me that these two values are very different.  

We agree. But there was no statistical difference between both treatments used. This is why we reached this conclusion.

Line 264: It is unclear how you derived these two values here. Elaborate so that the reader can understand. 

We clarified these two values in the discussion section

Line 281: change ‘from grapevine to grapevine’ to ‘between grapevines’

It has been corrected.

Line 276-278: Evidence is not convincing. Given the high prevalence of both viruses, it is expected that a large number of plants would be co-infected with both viruses. 

We agree with this appreciation, notice that we indicate that future epidemiological studies on GFkV are required to improve our knowledge of this virus and its putative vectors. We did not pretend to show evidence, only a possible explanation.

Line 287: change ‘grape composition’ to ‘the chemical composition of grape berries’

It has been corrected.

Line 302: change ‘dangerous’ to ‘damaging’;

It has been corrected.

Line 322: change ‘Contrariwise’ to ‘On the contrary’

After Referee 2 appreciation, the sentence was removed.

Line 333: change ‘big explants’ to ‘bigger explants’

It has been corrected.

Line 329: change ‘elimination resulted in easier than GFkV’ to ‘is easier to eliminate than GFkV’

It has been corrected.

Line 339: insert ‘viruses either’ after ‘highly infected with’

It has been corrected.

Supplementary Table S1: Divide the data into two columns, one for GLRaV-1 and the other GLRaV-3.

This data is presented in Table 2 (simple infection). Because the incidence of GLRaV-1 alone was detected only in cv. Malvasia tested plants. In general, GLRaV-1 was the least common and combined with other viruses.

Reviewer 2 Report

The manuscript: Recovering ancient grapevine cultivars in Balearic Islands: sanitary status evaluation and virus elimination (ID: plants-1773877) concerns the sanitary state of vines maintained in the germplasm collection of the Balearic Island and obtaining virus free plants.

The presented work is not innovative but provides knowledge about that which grapevine viruses are present in the region and what is their phytosanitary condition. The manuscript describes also the effectiveness of obtaining healthy plants but only within two out 33 tested cultivars which made the research preliminary. Moreover, the results concerns local grapevine cultivars which means they have little global significance.

However, generally the manuscript is well written. I have some comments which will help the authors to improve the manuscript to make it more accessible to a reader.

The authors should use cv. or cvs before (not after) the name of cultivars. This comment concerns the lines: 22, 314, 322.

Line 95: There is: …cultivars from this region (Argamussa and Gorgollassa)… while it would be more clearer: …cultivars (Argamussa and Gorgollassa) from this region …

The authors use the term “Germplasm Collection of Balearic Islands”. Is the Germplasm Collection of Balearic Islands formally recognized and financed by i.e. government? Or is it just collection of vines gathered by the reserchers’ group and localized at the experimental station? A capitalized name “Germplasm Collection of Balearic Islands” looks as if it was the proper name of the collection. Is it right?

Table 1 shows the results and should be presented in the section Results. However, the authors should list the tested cultivars describing plant material in Materials and methods.

In table 1 in the second column there is “No infected” while it should be “No infected (%)”.

Line 114: There is “Kits” while it should be “kits”.

Line 117: The authors should give the author of the reference, I mean it should be: … according to Padilla [43]… The same comment concerns line 257 (…reported by Velasco et al. [51].) and 319 (…Bota et al. [40] showed…).

Line 146: The authors wrote: “Real-time PCR analysis” but in fact the description concerns rather PCR (not real-time PCR). If it is PCR, the first sentence of the paragraph should be complete in information about reaction buffer and MgCl2 which probably were used.

Line 156 and 222, the caption under the figure 1. The authors used the word “varieties”. They should be consistent and use ”cultivar” in all manuscript.

It would be better to use different colours in figure 1, i.e. different hues of a colour for double infections, different hues of another colour for triple infections etc. The interpretation of the data would be easier.

The title of the Y axis is not clear enough. % of what? Of tested plants? The same is unclear in the caption o the figure.

Line 223. There is: “…varieties, all the plants analyzed had the same type of infection, such as Argamussa, Calop Blanc, Calop Negre and Calop Roig that were 100% infected with GLRaV-3 and GFkV…” 100% of plants within a cultivar? The authors should rephrase the sentence and specify. The same comment concerns the whole paragraph.

Line 309-311. Rephase  the sentence to be more clear: “The high GFLV elimination efficiency observed in Gorgollassa is also in line with that reached in “Bidaneh Sefid’ and ‘Shahroodi” cvs in Iran, combining meristem culture and thermotherapy [15].” It should be also: cvs Bidaneh Sefid and Shahroodi.

Author Response

The authors should use cv. or cvs before (not after) the name of cultivars. This comment concerns the lines: 22, 314, 322.

It has been corrected here and throughout the text.

Line 95: There is: …cultivars from this region (Argamussa and Gorgollassa)… while it would be more clearer: …cultivars (Argamussa and Gorgollassa) from this region …

It has been modified.

The authors use the term “Germplasm Collection of Balearic Islands”. Is the Germplasm Collection of Balearic Islands formally recognized and financed by i.e. government? Or is it just collection of vines gathered by the reserchers’ group and localized at the experimental station? A capitalized name “Germplasm Collection of Balearic Islands” looks as if it was the proper name of the collection. Is it right?

Yes, it is the official grapevine germplasm collection from the local government. The official name is Grapevine Germplasm Bank of Balearic Islands. We did the proper clarification in the text.   

Table 1 shows the results and should be presented in the section Results. However, the authors should list the tested cultivars describing plant material in Materials and methods.

We added a new table for M&M (Table 1). Now the old table 1 is table 2.

In table 1 in the second column there is “No infected” while it should be “No infected (%)”.

It has been corrected.

Line 114: There is “Kits” while it should be “kits”.

It has been corrected.

Line 117: The authors should give the author of the reference, I mean it should be: … according to Padilla [43]… The same comment concerns line 257 (…reported by Velasco et al. [51].) and 319 (…Bota et al. [40] showed…).

We added the authors of the references

Line 146: The authors wrote: “Real-time PCR analysis” but in fact the description concerns rather PCR (not real-time PCR). If it is PCR, the first sentence of the paragraph should be complete in information about reaction buffer and MgCl2 which probably were used.

The missing data has been added

Line 156 and 222, the caption under the figure 1. The authors used the word “varieties”. They should be consistent and use ”cultivar” in all manuscript.

It has been corrected throughout the ms

It would be better to use different colours in figure 1, i.e. different hues of a colour for double infections, different hues of another colour for triple infections etc. The interpretation of the data would be easier. The title of the Y axis is not clear enough. % of what? Of tested plants? The same is unclear in the caption o the figure.

Figure 1 is now in colour and the axes title modified

Line 223. There is: “…varieties, all the plants analyzed had the same type of infection, such as Argamussa, Calop Blanc, Calop Negre and Calop Roig that were 100% infected with GLRaV-3 and GFkV…” 100% of plants within a cultivar? The authors should rephrase the sentence and specify. The same comment concerns the whole paragraph.

It has been modified

Line 309-311. Rephase the sentence to be more clear: “The high GFLV elimination efficiency observed in Gorgollassa is also in line with that reached in “Bidaneh Sefid’ and ‘Shahroodi” cvs in Iran, combining meristem culture and thermotherapy [15].” It should be also: cvs Bidaneh Sefid and Shahroodi.

The sentence has been rewritten

Reviewer 3 Report

The manuscript requires a moderate level of revision prior to publication. For more details, see the attachment.

Author Response

The authors really appreciate Reviewer #3 evaluation of the first version of the manuscript. We have addressed all proposed changes to the text, additionally, some comments are addressed below.

Line 76. meristem cultrures and somatic embryogenesis is mostly used alone, and shoot tip cultures are generally combined with chemotherapy, cryotherapy or heattherapy.

We appreciate Reviewer#2 appreciation and we modified the sentence

Line 95: add 3-4 lines about the importance of these cultivars.

The recovery of old grapevine material not only contributes to preserve biodiversity but is a source of solutions to deal with the countless challenges that winegrowers face. The recovery of these two pre-phylloxera cultivars may ensure the use of traditional material rather than the massive planting of clones of widespread cultivars. The new reality of the sector is demanding traditional cultivars. The studies about the oenological potential of grapevine cultivars included in the germplasm bank reported very good results for these two cultivars (Garcia-Muñoz, 2011 (PhD thesis)). These two Minor varieties excluded from Quality Demarcations appear to be more aromatic and, as a consequence, Argamusa, Gorgollassa varieties could be taken into consideration for the development of new wine market strategies, which could also play an important role in the conservation of these cultivars.

A short explanation was included in the text (pg 2, lines 54-58)

Line 133: Did the phenotype of treated plants change after sanitation?

No significant changes in plant performance were appreciated after sanitation. Actually. Gorgollassa is already authorized and under cultivation.

Line 146: real-time or reverese transcription? It seems a normal PCR amplification with gel eletrophoresis

The mistake has been corrected.

Line 158: In perlite? Which method was used for rooting? In a dark chamber or in the greenhouse?

The missing information was added to the M&M section (pg 4, lines 172-174)

Line 161: I suggest presenting pictures about sanitation procedures from the rooted cuttings to the rooted in vitro/greenhouse plants

We added the pictures as supplemental material

Line 183: Then plants were not isolated form insects and nematods. Have been these virus-free plants tested by local authorities?

The plants were tested again prior to authorization by the national government.

Line 188: Which software was used? Please, insert the statistical analysis in the supplementary material

As indicated in the next sentence, we used JMP 10 software package. We clarified this in the text. We include the statistical data as S3

Line 232: If Gorgollassa was virus free, why it was necessary to sanitize it?

As indicated in the text Gorgollassa showed double (GFLV, GLRaV-4-9) and triple (GLRaV-3, GFLV, GLRaV-4-9) infections.

Line 235-243: Please, insert a table into the supplementary material, when readers can find the ELISA and RT-PCR results all of the mother and aclimatized plant!

Please, insert additional  figures about gel electrophoresis of PCR amplified virus fragments of the mother plants

Authors appreciate Reviewer #2 suggestion, but we consider it a low utility to include all raw data in this kind of studies. Noted that the ELISA results (absorbances) should be considered a qualitative parameter. On the other hand, the PCR results are extracted from the analysis of the gel electrophoresis to indicate the presence/absence of the virus. To present a supplementary table with all the results including mother plants and in vitro obtained ones, with the information on the presence/absence of each virus makes no sense to us. We really believe that all these results are well represented in the tables and figures and to include the raw data will not add valuable information. In the same line, it is not possible to include all the gel electrophoresis pictures of all the plants tested. We can include one as an example but we certainly believe that no additional information will be added to the manuscript.

Line 309: This is very surprising, because GFLV is non-phloem-limited virus, and it can spread even in the meristemic tissues. Please, give a hypothesis, how could be successful the elinination process using 1-3 mm long shoot tips

Our results demonstrate that even with a non-phloem limited virus this combination of thermotherapy with shoot tips is effective in a %. Our hypothesis is that the virus replication was limited by the high temperature.

Line 323: It is surprising, because GFKV is a highly phloem-limited virus, and in my experience, it is easy to eliminate this virus from other cultivars using meristem cultures. Please, give some references about the recalcitrance to sanitation of GFKV

We agree with this appreciation and our intention was not to suggest recalcitrance to the sanitation of GFKV. We tried to clarify in the text.

Line 324: I did not find this part in the cited review about GFKV. I could not find this article. Please, give some lines about the used sanitation and diagnostics methods, and some data about the effectiveness.  There are other articles about this phenomenon?

According to these appreciations, we revised the literature and rewrite these statements.

Line 324: I think, that the maximum size of excised meristems is 0.5-0,7 mm. 1 mm long shoot tips are not called meristems. Please, check in Panattoni et al 2013, or in other reliable sources. Maybe: microshoot culture

We called meristem because the authors called meristems and indicate the 1 mm size, but as we agree with the referee we change “meristem” for “microshoot culture”.
